# Combined effect of the entomopathogenic fungus *Metarhizium robertsii* and avermectins on the survival and immune response of *Aedes aegypti* larvae

Yuriy A. Noskov[1,2], Olga V. Polenogova[1], Olga N. Yaroslavtseva[1], Olga E. Belevich[1], Yuriy A. Yurchenko[1], Ekaterina A. Chertkova[1], Natalya A. Kryukova[1], Vadim Yu Kryukov[1] and Viktor V. Glupov[1]

[1] Institute of Systematics and Ecology of Animals, Siberian Branch of Russian Academy of Sciences, Novosibirsk, Russia
[2] Tomsk State University, Tomsk, Russia

Corresponding author
Yuriy A. Noskov,
yunoskov@gmail.com

## ABSTRACT

Combination of insect pathogenic fungi and microbial metabolites is a prospective method for mosquito control. The effect of the entomopathogenic fungus *Metarhizium robertsii* J.F. Bischoff, S.A. Rehner & Humber and avermectins on the survival and physiological parameters of *Aedes aegypti* (Linnaeus, 1762) larvae (dopamine concentration, glutathione S-transferase (GST), nonspecific esterases (EST), acid proteases, lysozyme-like, phenoloxidase (PO) activities) was studied. It is shown that the combination of these agents leads to a synergistic effect on mosquito mortality. Colonization of *Ae. aegypti* larvae by hyphal bodies following water inoculation with conidia is shown for the first time. The larvae affected by fungi are characterized by a decrease in PO and dopamine levels. In the initial stages of toxicosis and/or fungal infection (12 h posttreatment), increases in the activity of insect detoxifying enzymes (GST and EST) and acid proteases are observed after monotreatments, and these increases are suppressed after combined treatment with the fungus and avermectins. Lysozyme-like activity is also most strongly suppressed under combined treatment with the fungus and avermectins in the early stages posttreatment (12 h). Forty-eight hours posttreatment, we observe increases in GST, EST, acid proteases, and lysozyme-like activities under the influence of the fungus and/or avermectins. The larvae affected by avermectins accumulate lower levels of conidia than avermectin-free larvae. On the other hand, a burst of bacterial CFUs is observed under treatment with both the fungus and avermectins. We suggest that disturbance of the responses of the immune and detoxifying systems under the combined treatment and the development of opportunistic bacteria may be among the causes of the synergistic effect.

## INTRODUCTION

Mosquitoes are obligate intermediate hosts for a variety of pathogens that cause human mortality and morbidity worldwide. *Aedes aegypti* is considered to be an important vector

of human diseases such as dengue and yellow fever, chikungunya, and Zika infections (*Tolle, 2009*; *Bhatt et al., 2013*), and its control is therefore an objective to prevent the transmission of these diseases. Chemical insecticides are still the most important element in mosquito control programs, despite direct and indirect toxic effects on nontarget organisms, including humans. In addition, chemicals induce resistance in a number of vector species (*Vontas, Ranson & Alphey, 2010*; *Ranson & Lissenden, 2016*; *Smith, Kasai & Scott, 2016*). Therefore, there is a need for alternative nonchemical vector control approaches. Classical biological control based on using various microorganisms, such as entomopathogenic fungi and bacteria, is a frequent tool for addressing this issue.

Among the biological agents employed for mosquito larvae control, bacteria from the genus *Bacillus* are the most widely used. In addition, products of the entomophatogenic fungi *Metarhizium anisopliae s.l.*, and *Beauveria bassiana s.l.* are actively being developed for use against mosquito adults and larvae (*Butt et al., 2013*; *Greenfield et al., 2015*; *Ortiz-Urquiza, Luo & Keyhani, 2015*). It should be noted that mosquitoes and other insects can develop resistance to the *Bacillus thuringiensis* Berliner biological larvicide (*Tilquin et al., 2008*; *Paris et al., 2011*; *Boyer et al., 2012*). However, the resistance of insects to entomopathogenic fungi develops very slowly (*Dubovskiy et al., 2013*). Various species of mosquito larvae present different susceptibilities to *Metarhizium*, among which *Ae. aegypti* is the least susceptible (*Greenfield et al., 2015*; *Garrido-Jurado et al., 2016*). Thus, a concentration of conidia that is effective for *Ae. aegypti* control would affect a range of nontarget aquatic invertebrates. Recent studies have found that some nontarget aquatic species are more sensitive to fungal metabolites (*Garrido-Jurado et al., 2016*) and conidia (*Belevich et al., 2017*) than target mosquito species. To reduce toxic effects on the aquatic environment and increase efficacy against mosquitoes, entomopathogenic fungi may be combined with other biocontrol agents or low doses of natural insecticides. For example, combined treatment with *Metarhizium* and mosquito predator species (*Toxorhynchites*) has shown additive or synergistic effects on the mortality of *Ae. aegypti* (*Alkhaibari et al., 2018*). However, few studies have been carried out to determine the effect of combined treatment with entomopathogenic fungi and other insecticides or plant or microbial metabolites as a potential tool for improving mosquito larvae control. Synergistic effects between entomopathogenic fungi and some chemical insecticides (temephos, spinosad) (*Shoukat et al., 2018*; *Vivekanandhan et al., 2018*) or biological agents (*Azadirachta indica*) A. Juss (*Badiane et al., 2017*) on the mortality of mosquito larvae have been found. However, the physiological and biochemical aspects of this synergism were not considered.

One type of promising insecticide that can be effectively used for mosquito vector control is the avermectins. Avermectins are a class of macrocyclic lactones isolated from the soil actinomycete *Streptomices avermitilis* (ex Burg et al.) Kim and Goodfellow (*Drinyaev et al., 1999*) and include several commercial derivatives (ivermectin, abamectin, doramectin and eprinomectin) with the same mode of action—activation of glutamate-gated chloride channels, followed by uncontrolled influx of chloride ions into the cells, which leads to paralysis and death of the organism (*Campbell et al., 1983*). At the same time, avermectins are relatively safe for humans (*Crump & Omura, 2011*). Previous studies have shown that avermectins are efficient for the control of *Culex quinquefasciatus* Say (*Freitas et al., 1996*;

*Alves et al., 2004*), *Anopheles albimanus*, Wiedemann *An. stephensi* Liston (*Dreyer, Morin & Vaughan, 2018*), and *An. gambiae* Giles (*Alout et al., 2014*; *Chaccour et al., 2017*). However, most of these studies have been carried out with adult mosquitoes feeding on blood containing ivermectin. Avermectins exhibit a relatively short half-life period, which limits their ability to kill mosquitoes but may be compensated by the application of multiple treatments or use of higher concentrations. However, these approaches may contribute to the development of mosquito larvae resistance to avermectins (*Su et al., 2017*). We hypothesize that the interaction of entomopathogenic fungi with avermectins can have a stable insecticidal effect at relatively low concentrations and is a promising combination for safe and effective mosquito control.

It is important that entomopathogenic fungi such as *Metarhizium* are adapted to terrestrial hosts and that in mosquito larvae, the fungi do not adhere to the cuticle surface and do not germinate through integuments into the hemocoel. Conidia ingested by mosquito larvae do not penetrate the gut wall (*Butt et al., 2013*). Thus, a "classic" host-pathogen interaction does not occur, and larval mortality is associated with stress induced by spore-bound proteases on the surface of ingested conidia (*Butt et al., 2013*). These authors suggest that fungal proteases cause an increase in the activity of caspases in mosquitoes, which leads to apoptosis, autolysis of tissues and death of the larvae. The activation of detoxifying enzymes and antimicrobial peptides (AMPs) occurs in larvae infected with the fungus but is not sufficient to protect the larvae from death. As a rule, mosquito larvae die showing symptoms of bacterial decomposition after treatment with *Metarhizium* and *Beauveria* (*Scholte et al., 2004*). Therefore, this pathogenesis can be considered mixed (both bacterial and fungal).

During this process, particular superfamilies of enzymes such as glutathione-S-transferases (GST) and nonspecific esterases (EST) are usually involved in the biochemical transformation of xenobiotics (*Li, Schuler & Berenbaum, 2007*). Various hormones such as biogenic amines are involved in insect stress reactions. Among them, the role of the neurotransmitter dopamine (which serves as a neurohormone as well) in this process remain poorly understood. It is known that dopamine mediates phagocytosis and is involved in the activation of the pro-phenoloxidase (proPO) cascade, thus playing an important role in fungal and bacterial pathogenesis as well as in the development of toxicoses caused by insecticides (*Delpuech, Frey & Carton, 1996*; *Gorman, An & Kanost, 2007*; *Wu et al., 2015*). In addition, both pathogens and toxicants can lead to changes in the antimicrobial activity of insects and the bacterial load that can affect the susceptibility of insects to pathogenic fungi (*Wei et al., 2017*; *Ramirez et al., 2018*; *Polenogova et al., 2019*). It should be noted that the above mentioned physiological reactions in mosquito larvae under the combined action of entomopathogenic fungi and insecticides have not yet been studied.

The aims of this study were (1) to determine the susceptibility of *Aedes aegypti* larvae to combined treatment with avermectins and *Metarhizium robertsii* and (2) estimate their immune and detoxificative responses to *M. robertsii* and avermectins either alone or their combination.

## MATERIALS & METHODS

### Insecticides and fungi

The entomopathogenic fungus *Metarhizium robertsii* (strain MB-1) from the collection of microorganisms of the Institute of Systematics and Ecology of Animals SB RAS was used in this work. The conidia of the fungus were grown on autoclaved millet for 10 days at 26 °C in the dark, followed by drying and sifting (*Belevich et al., 2017*). The industrial product "Phytoverm" 0.2% (SPC "Pharmbiomed", Russia) was used in these experiments and includes a complex of natural avermectins (A1a (9%), A2a (18%), B1a (46%), B2a (27%)) produced by *Streptomyces avermitilis*.

### Insect maintenance and toxicity tests

*Aedes aegypti* larvae from the collection of the Institute of Systematics and Ecology of Animals SB RAS were maintained in tap water in the laboratory at 24 °C (±1 °C) under a natural photoperiod (approximately 16:8 light:dark). The larvae were fed Tetramin Junior fish food (Tetra, Germany). The susceptibility of *Ae. aegypti* to both avermectins and the conidia of *M. robertsii* was tested in 200 ml plastic containers containing 100 ml of water with 15 larvae. Third-4th-instar larvae were used in the experiment. The experiment involved four treatments: control, fungus, avermectins, and fungus + avermectins. The fungal conidia and avermectins were suspended in distilled water, vortexed and applied separately or together to the containers with mosquito larvae at a volume of 2 ml per container. The final conidial concentration for infection was $1 \times 10^6$ conidia/ml. The final concentration of avermectins was 0.00001%/ml. The control was treated with the same amount of distilled water. Mortality was assessed daily for 6 days. Ten replicates with 15 larvae were performed for each treatment.

### Light microscopy and colonization assessment

Forty-eight hours posttreatment (pt), mosquito larvae ($n = 3$ for each treatment) were collected in 2% glutaraldehyde containing 0.1 M Na-cacodylate buffer (pH 7.2) and were maintained at 4 °C for 1–24 h. Semi-thin sections were stained with crystal violet and basic fuchsin and were observed with a phase-contrast microscope (Axioskop 40, Carl Zeiss, Germany).

To assess fungal colonization, 48 h pt and newly dead larvae (4–6 days pt) were cut open, and their internal contents were squeezed onto a glass side. The contents were examined for the presence/absence of hyphal bodies using light microscopy ($n = 30$ for each treatment). Newly dead larvae were placed on moistened filter paper in Petri dishes ($n = 30$) to determine the germination and surface sporulation of *Metarhizium*.

### Total larval body supernatant

Mosquito larvae bodies of individual 4th-instar larvae of *Ae. aegypti* were collected in 50 μl of cool (+4 °C) 0.01 M PBS (50 mM, pH 7.4, 150 mM NaCl) with 0.1 mM N-phenylthiourea (PTU) to measure GST, EST and acid protease activities or without PTU to measure phenoloxidase (PO) activity. Then, the samples were sonicated in an ice bath with three 10 s bursts using a Bandelin Sonopuls sonicator. The sample solution was centrifuged

at 20.000 g for 5 min at +4 °C. The obtained supernatant was directly used to determine enzyme activities.

## Detection of phenoloxidase, glutathione-S-transferase and esterase activity

The activities of PO, GST and EST were measured at 12 and 48 h after exposure ($n = 20$ per treatment for each enzyme).

PO activity was assayed by using a method modified from that described by *Ashida & Söderhäll (1984)*. The PO activity of the larval homogenates was determined spectrophotometrically on the basis of the formation of dopachrome at a wavelength of 490 nm. Aliquots of the samples (10 µl) were added to microplate wells containing 200 µl of 10 mM 3.4-dihydroxyphenylalanine and incubated at 28 °C in the dark for 45 min. The PO activity was measured kinetically every 5 min and the time point was chosen according to Michaelis constant.

The activity of EST was measured using the method of *Prabhakaran & Kamble (1995)* with some modifications. Aliquots of the samples (3 µl) were added to microplate wells containing 200 µl of 0.01% p-nitrophenylacetate and incubated for 10 min at 28 °C. The activity of EST was determined spectrophotometrically at a wavelength of 410 nm on the basis of the formation of nitrophenyl. The EST activity was measured kinetically every 2 min and the time point was chosen according to Michaelis constant.

The measurement of GST activity was carried out according to the method of *Habig, Pabst & Jakoby (1974)* with some modifications. Aliquots of the samples (7 µl) were added to microplate wells containing 200 µl of 1 mM glutathione and 5 µl of 1 mM 2.4-Dinitrochlorobenzene and incubated at 28°C for 12 min. The activity of GST was determined spectrophotometrically on the basis of the formation of 5-(2.4-dinitrophenyl)-glutathione at a 340 nm wavelength. The GST activity was measured kinetically every 3 min and the time point was chosen according to Michaelis constant.

Enzymes activity was measured in units of the transmission density ($\Delta$A) of the incubation mixture during the reaction per 1 min and 1mg of protein. The protein concentration in the samples was determined by the method of *Bradford (1976)*. To generate the calibration curve, bovine serum albumin was used.

## Dopamine concentration measurements

Dopamine concentrations were measured at the 12 and 48 h pt in individual larval bodies ($n = 10$ per treatment). Mosquito larvae were homogenized in 30 µl of phosphate buffer and incubated in a Biosan TS 100 Thermoshaker for 10 min at 28 °C and 600 rpm, then incubated at room temperature for 20 min and centrifuged at 4 °C and 10.000 g for 10 min. The supernatants were transferred to clean tubes and centrifuged with the same settings for 5 min. Before transfer to the chromatograph, the samples were filtered.

Dopamine concentrations were measured by an external standard method using an Agilent 1,260 Infinity high-performance liquid chromatograph with an EsaCoulochem III electrochemical detector (cell model 5010A, potential 300 mV) according to the method of *Gruntenko et al. (2005)* with some modifications. Dopamine hydrochloride

(Sigma-Aldrich) was used as a standard. Separation was performed in a ZorbaxSB-C18 column (4.6–250 mm, particles 5 μm) in isocratic mode. Mobile phase: 90% buffer (200 mg/l 1-OctaneSulfonicAcid (Sigma-Aldrich), 3.5 g/l KH$_2$PO$_4$) and 10% acetonitrile. The flow rate was 1 ml/min. Chromatogram processing was performed using ChemStation software, and the amount of dopamine was determined by comparing the peak areas of the standard and the sample.

### Acid proteases

Acid protease activity was measured using method described by *Anson (1938)* with modifications. Fifty μl of the homogenate supernatant was added to 250 μl of 0.1 M acetate buffer (pH 4.6) containing 0.3% hemoglobin (Sigma, CAS number 9008-02-0). The samples were incubated for 60 min at 27 °C, and the reaction was stopped by adding 500 μl of 5% TCA and cooling on ice for 10 min at 4 °C. The samples were centrifuged at 14,000 g for 5 min at 4 °C, and the enzyme activity was determined spectrophotometrically at a wavelength of 280 nm in a 96-well plate reader.

### Lysozyme-like activity

Lysozyme-like activity in the mosquito homogenate was determined through analysis of the lytic zone by diffusion into agar. Ten milliliters of Nutrient Agar (NA) (HiMedia, India) and *Micrococcus lysodeikticus* bacteria ($1 \times 10^7$ cells/ml) were added to Petri dishes. The agar was perforated to create 2 mm-diameter wells, which were then filled with 3 μl of full-body homogenate, followed by incubation at 37 °C for 24 h. Series of dilutions of chicken egg white lysozyme (EWL) (Sigma) (0.5 mg/ml, 0.2 mg/ml, 0.1 mg/ml, 0.005 mg/ml, 0.001 mg/ml) were added to each dish, allowing us to obtain a calibration curve based on these standards. Lytic activity was determined by measuring the diameter of the clear zone around each well and expressed as the equivalent of EWL (mg/ml) (*Mohrig & Messner, 1968*).

### CFU counts of Metarhizium and cultivated bacteria in infected larvae

Homogenates of the mosquitoes (3 larvae per sample) were suspended in 1 ml of sterile aqueous Tween-20 (0.03%), and the suspensions were then diluted 50-fold. Next, 100 μl aliquots were inoculated onto the surface of modified Sabouraud agar (10 g peptone, 40 g D-glucose anhydrous, 20 g agar, 1 g yeast extract) supplemented with an antibiotic cocktail (acetyltrimethyl ammonium bromide 0.35 g/L; cycloheximide 0.05 g/L; tetracycline 0.05 g/L; streptomycin 0.6 g/L; PanReacAppliChem, Germany) for the inhibition of bacteria and saprotrophic fungi. The Petri dishes were maintained at 28 °C in the dark. The colonies were then counted after 7 days.

For the estimation of cultivated bacterial CFU counts, homogenates of larvae (3 larvae per sample) were suspended in 1 ml of 0.1 M phosphate buffer. Then, the suspension was diluted to $10^{-2}$, $10^{-3}$ and $10^{-4}$. Aliquots of 100 μl of the larval dilutions were inoculated onto the surface of blood agar media (HiMedia, Mumbai, India). The Petri dishes were maintained at 28 °C. The colonies were counted after 48 h. Three samples of each treatment were used in the analysis.

## Statistical analysis

Data were analyzed using GraphPad Prism v.4.0 (GraphPad Software Inc., USA), Statistica 8 (StatSoft Inc., USA), PAST 3 (*Hammer, Harper & Ryan, 2001*) and AtteStat 12.5 (*Gaidyshev, 2004*). Differences between synergistic and additive effects were determined by comparing the expected and observed insect mortality using the $\chi^2$ criterion (*Robertson & Preisler, 1992*). The expected mortality from dual treatment was calculated by the formula $P_E = P_0 + (1 - P_0) \times (P_1) + (1 - P_0) \times (1 - P_1) \times (P_2)$, where $P_E$ is the expected mortality after combined treatment with fungus and avermectins, $P_0$ is mortality in the control groups, $P_1$ is the mortality posttreatment with *M. robertsii*, $P_2$ is the mortality posttreatment with avermectins. The $\chi^2$ values were calculated by the formula $\chi^2 = (L_0 - L_E)^2/L_E + (D_0 - D_E)^2/D_E$, where $L_0$ is the observed number of survived larvae, $L_E$ is the expected number of surviving larvae, $D_0$ is the observed number of dead larvae, and $D_E$ is the expected number of dead larvae. This formula was used to test the hypothesis of independence (1 df: $P = 0.05$). Additive effect was indicated if $\chi^2 < 3.84$. A synergistic effect was indicated if $\chi^2 > 3.84$ and observed mortality greater than the expected one. A value of 3.84 corresponds to $P < 0.05$ with a degree of freedom = 1. The Kaplan–Meier test was used to calculate the median lethal time (presented as LT50 ± SE). A log-rank test was used to quantify differences in mortality dynamics. As the distribution of the physiological parameters except for the dopamine concentration deviated from a normal distribution (Shapiro–Wilk test, $P < 0.05$), we used the nonparametric equivalent of a two-way ANOVA: the Scheirer-Ray-Hare test (*Scheirer, Ray & Hare, 1976*), followed by Dunn's post hoc test. The data on dopamine concentrations passed the normality test (Shapiro–Wilk test, $P > 0.05$) and were analyzed by two-way ANOVA followed by Tukey's post hoc test. Differences between *Metarhizium* CFU counts were compared by $t$-tests.

# RESULTS

## Synergy between avermectins and the fungus

Significant differences in the dynamics of larval mortality between the treatments were observed (log-rank test: $\chi^2 = 397.3$, $df = 3$, $P < 0.0001$; Fig. 1). Treatment with avermectins or conidia of *M. robertsii* led to 57 and 55% mortality, respectively, whereas combined treatment led to 99% mortality at the 6th day pt. Mortality in the control treatment did not exceed 1%. The median lethal time post-combined treatment (3 ± 0.1 d) occurred twice as fast as under treatment with avermectins (6 ± 0.3 d) or the fungus (6 d ± inf.) alone ($\chi^2 > 113.7$, $df = 1$, $P < 0.0001$).

From the 2nd to the 6th day pt, the avermectins and fungus interacted synergistically ($\chi^2 > 18.5$, $df = 1$, $P < 0.001$, ESM Table S1). These effects were consistently observed in four independent experiments.

## Colonization assay

At 48 h pt, we observed mass accumulation of *Metarhizium* conidia in the gut lumen (Fig. 2A). Germinated conidia were not detected in larvae at 48 h pt ($n = 12$). However, in one sample (combined treatment), hyphal bodies were detected in the hemocoel (Fig. 2A). In the newly dead larvae after the fungal and combined treatments (4–6 days), we detected

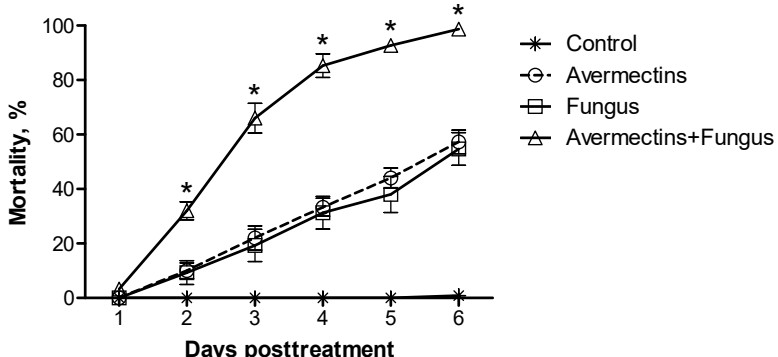

**Figure 1** **Mortality dynamics of *Ae. aegypti* larvae after treatment with *M. robertsii* ($1 \times 10^6$ conidia/ml), avermectins (0.00001%) and their combination.** The control was treated with distilled water. The asterisks (*) indicate a synergistic effect ($\chi^2 > 18.5$, $df = 1$, $P < 0.001$, see Table S1).

colonization of the hemocoel with hyphal bodies (Fig. 2B). Under the combined treatment, 83% hypha-positive larvae were found, while in the fungal treatment, 90% hypha-positive larvae were recorded. No significant differences between these treatments were observed ($\chi^2 = 0.58$, $df = 1$, $P = 0.45$, $n = 30$ larvae per treatment). No hyphal bodies were detected in the fungus-free treatments. A total of 70% and 60% of larvae were overgrown with *Metarhizium* under incubation in moist chambers (Fig. 2C) after treatment with the fungus or the mixture (avermectins + fungus), respectively. Only nongerminated conidia, but no hyphal bodies, were detected in the water in which treated larvae were maintained (Fig. 2D).

## Phenoloxidase activity

At 12 h pt, we registered a significant decrease in PO activity under the influence of fungal infection (Scheirer-Ray-Hare test, effect of fungus: $H_{1.52} = 12.6$, $P = 0.00038$; Fig. 3). Avermectins did not significantly change PO activity ($H_{1.52} = 0.3$, $P = 0.54$). A stronger decrease in enzyme activity was observed after combined treatment, but a significant factor interaction was not revealed ($H_{1.52} = 1.9$, $P = 0.16$). At 48 h pt, we detected a significant increase in PO activity under the influence of avermectins ($H_{1.32} = 5.33$, $P = 0.02$). The effect of the fungus was not significant ($H_{1.32} = 0.7$, $P = 0.39$), but a tendency toward an interaction between the factors was revealed ($H_{1.32} = 3.2$, $P = 0.07$). This is explained by the inhibition of PO activity by the fungus alone (Dunn's test, $P = 0.01$, $P = 0.04$, compared to the control and avermectin treatments, respectively) and by the tendency of increased enzyme activity after combined treatment.

## Dopamine concentration

The effects of the fungus or avermectins on the dopamine concentration at 12 h pt were not significant ($F_{1.31} = 1.2$, $P = 0.27$; Fig. 4), although a trend toward a factor interaction was revealed ($F_{1.31} = 3.3$, $P = 0.07$). This was due to a clear tendency to decrease the dopamine concentration after treatment with the fungus alone (HSD Tukey test, $P = 0.07$ compared to fungus-free treatments) but not with the combination of the fungus and avermectins

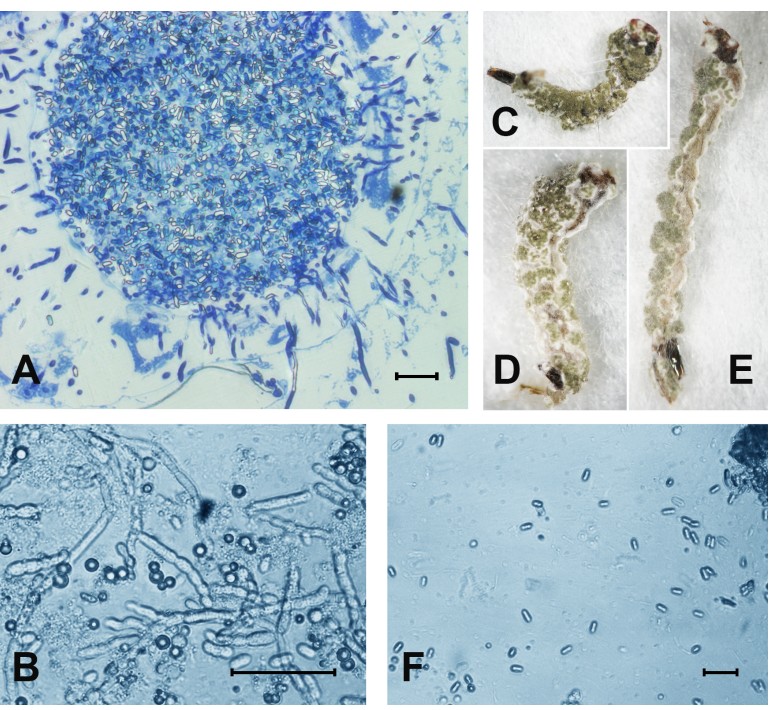

**Figure 2** **The colonization of *Ae. aegypti* by *M. robertsii*.** (A) Accumulation of conidia in the gut and colonization of the hemocoel by hyphal bodies. (B) Colonization of the fat body. (C–E) Mosquito larvae with surface conidiation of *Metarhizium* in a moist chamber. (F) Nongerminated conidia in a sample of water in which infected larvae were maintained. Scale bar: 20 μm.

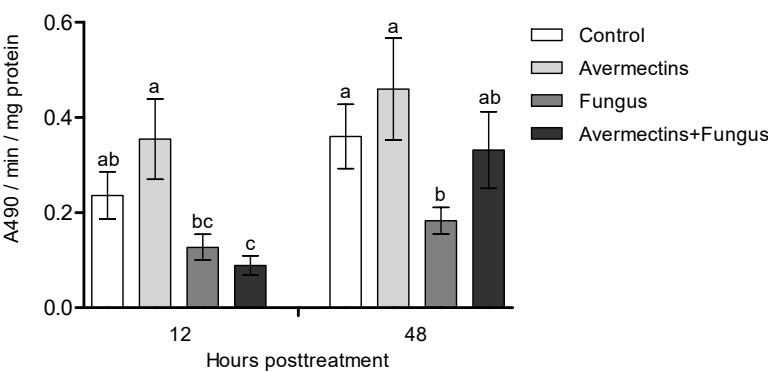

**Figure 3** **Activity of PO in the whole-body homogenates of *Ae. aegypti* larvae after treatment with *M. robertsii*, avermectins and their combination.** In the control treatment, equal amounts of water were added. Error bars represent the standard error of the mean. Significant differences are indicated with different letters within one time point (Dunn's test, $P < 0.05$).

($P = 0.47$ compared to fungus-free treatments). At 48 h pt, we observed a significant decrease in the dopamine concentration under the influence of the fungus ($F_{1.31} = 4.62$, $P = 0.03$); however, there were no significant differences between the treatments (HSD

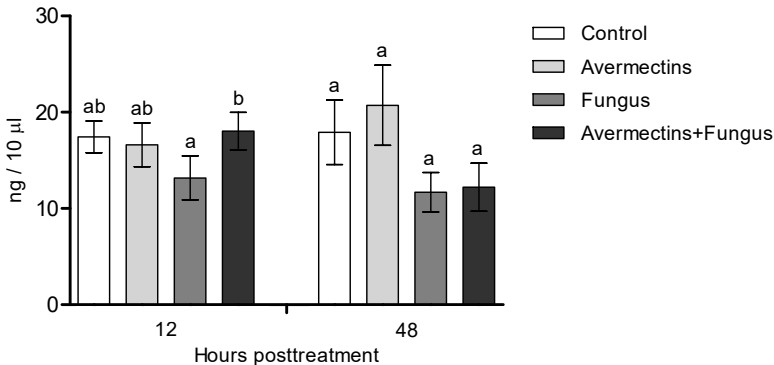

**Figure 4 Dopamine concentration in whole-body homogenates of *Ae. aegypti* larvae after treatment with *M. robertsii*, avermectins and their combination.** In the control treatment, equal amounts of water were added. Error bars represent the standard error of the mean. Significant differences are indicated by different letters within one time point (HSD Tukey test, $P < 0.05$).

Tukey test, $p = 0.1$). No significant interaction effects between the factors on the dopamine concentration at 48 h pt were detected ($F_{1.31} = 0.0$, $P = 1.0$).

## Detoxifying enzymes

At 12 h pt, an interaction effect between the two factors (avermectins and the fungus) on GST activity was observed ($H_{1.44} = 8.1$, $P = 0.0043$, Fig. 5A). Avermectins and the fungus alone significantly (1.5–2-fold) increased GST activity compared to untreated larvae (Dunn's test, $P = 0.007$, $P = 0.001$, respectively), but after combined treatment, the enzyme activity did not significantly differ from that in the control. Similar patterns were registered for EST activity at 12 h pt (Fig. 5B). In this case, EST was activated under the influence of avermectins alone (Dunn's test, $P = 0.002$, compared to control), but fungal infection inhibited this activation. In particular, EST activity in the fungal and combined treatments did not differ from that in the control (Dunn's test, $P = 0.28$, $P = 0.34$, respectively).

At 48 h pt, we observed a significant increase in GST activity under the influence of avermectins ($H_{1.32} = 4.2$, $P = 0.03$). The effect of the fungus as well as the interaction between the factors on GST activity at this time point was not significant ($H_{1.32} = 2.5$, $P = 0.1$ and $H_{1.32} = 0.61$, $P = 0.43$, respectively). EST activity nonsignificantly increased under the influence of avermectins ($H_{1.32} = 1.85$, $P = 0.17$). The effect of the fungus on enzyme activity was not significant ($H_{1.32} = 0.001$, $P = 0.96$), and no significant interactions between the factors were detected ($H_{1.32} = 0.49$, $P = 0.48$).

## Acid protease activity

At 12 h pt, a significant interaction between avermectins and the fungus on acid protease activity was observed ($H_{1.48} = 14.8$; $P = 0.00011$; Fig. 6). In particular, protease activity was increased after fungal treatment alone (3-fold compared to control, $P = 0.0003$) but not after combined treatment. At 48 h pt, protease activity was strongly increased under the influence of avermectins ($H_{1.43} = 27.4$, $P = 0.000016$), and a trend toward an increase in

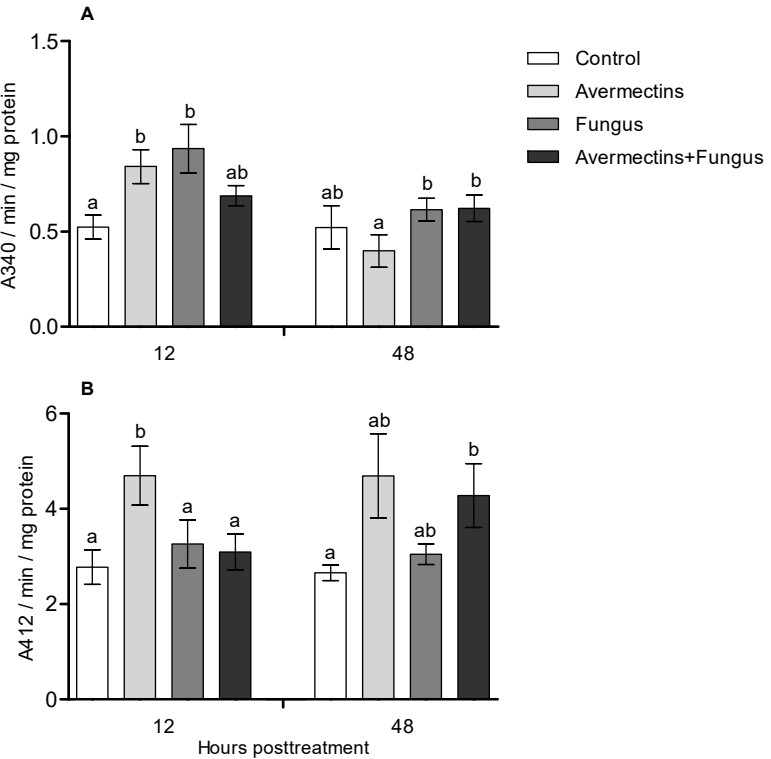

**Figure 5** **GST (A) and EST (B) activity in whole-body homogenates of *Ae. aegypti* larvae after treatment with *M. robertsii*, avermectins and their combination.** In the control treatment, equal amounts of water were added. Error bars represent the standard error of the mean. Significant differences are indicated by different letters within one time point (Dunn's test, $P < 0.05$).

enzyme activity was registered under the influence of the fungus ($H_{1.43} = 3.18$, $P = 0.07$). No significant interaction between the factors was revealed, although a trend toward the highest increase in protease activity was registered after the combined treatment.

## Lysozyme-like activity

At 12 h pt, we recorded a decrease in lysozyme-like activity under the influence of both fungal infection and avermectins (effect of fungus: $H_{1.56} = 6.46$, $P = 0.011$; effect of avermectins: $H_{1.56} = 14.04$, $P = 0.00017$; Fig. 7). The greatest decrease was observed after the combined treatment (Dunn's test, $P < 0.001$, compared with the other treatments). At 48 h pt, a sharp (1.7–2.0-fold) increase in lysozyme-like activity was recorded under the influence of avermectins ($H_{1.116} = 68.6$, $P < 0.0000001$). The effect of the fungus on the level of the enzyme at 48 h pt was not significant. No significant interaction effect between the factors on the level of lysozyme was observed at 12 and 48 h pt ($H = 0.23$, $P = 0.63$).

## Fungal and bacterial CFUs

The plating of mosquito larval homogenates on modified Sabouraud agar showed significant differences in the *Metarhizium* CFUs between treatment with the fungus either alone or combined with avermectins (Fig. 8A). The *Metarhizium* CFU count in the

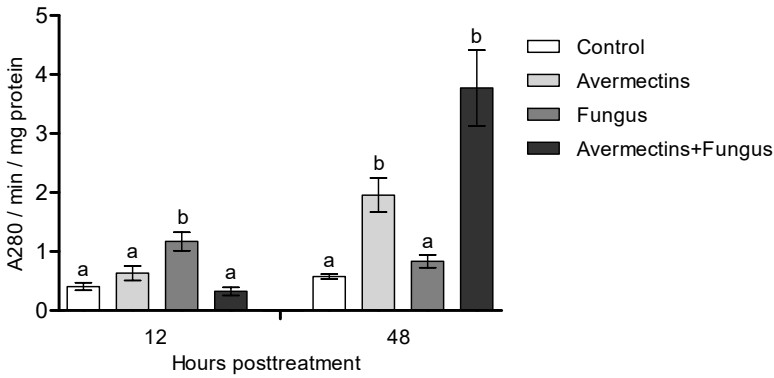

**Figure 6  Acid protease activity in whole-body homogenates of *Ae. aegypti* larvae after treatment with *M. robertsii*, avermectins and their combination.** In the control treatment, equal amounts of water were added. Error bars represent the standard error of the mean. Significant differences are indicated by different letters within one time point (Dunn's test, $P < 0.05$).

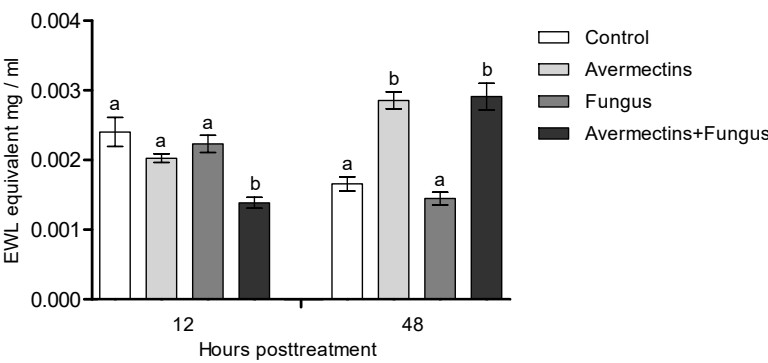

**Figure 7  Lysozyme-like activity in whole-body homogenates of *Ae. aegypti* larvae after treatment with *M. robertsii*, avermectins and their combination.** In the control treatment, equal amounts of water were added. Error bars represent the standard error of the mean. Significant differences are indicated by different letters within one time point (Dunn's test, $P < 0.05$).

fungal treatment was twice as high as that in the combined treatment ($t = 6.4$, $df = 18$, $P = 0.001$). Homogenates of the larvae from the fungus-free treatments (avermectin alone and control) did not form any fungal colonies.

The plating of larval homogenates on blood agar showed a significant (17–75-fold) increase in bacterial CFUs after treatment with the fungus and avermectins. A significant effect was registered for avermectins ($H_{1.19} = 4.8$, $P = 0.03$; Fig. 8B) but not for the fungus ($H_{1.19} = 1.9$, $P = 0.17$). However, a clear tendency toward an increase in CFUs was registered after treatment with the fungus alone (Dunn's test, $P = 0.054$, compared to control). No significant interaction effect between the fungus and avermectins on bacterial CFUs was revealed ($H_{1.19} = 1.9$, $P = 0.17$).

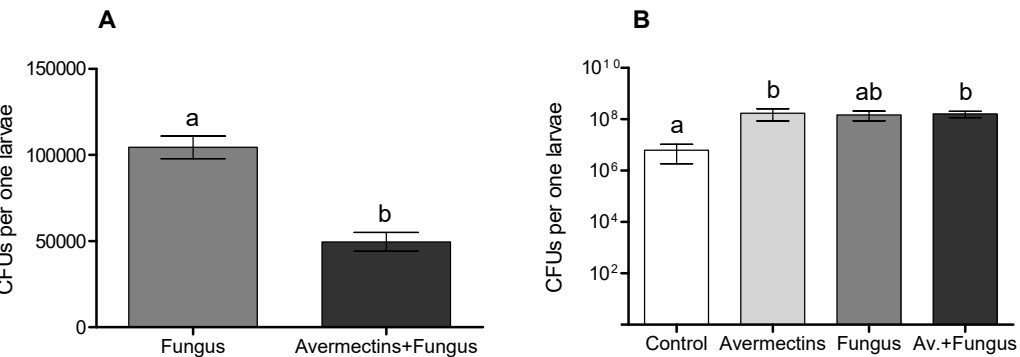

**Figure 8** **Colony forming units of *M. robertsii* (A) and cultivable bacteria (B) in whole-body homogenates of *Ae. aegypti* larvae after treatment with *M. robertsii*, avermectins and their combination.** Error bars show min and max values. Significant differences are indicated by different letters (*t*-test, $P < 0.001$, for fungal CFU, and Dunn's test, $P < 0.05$, for bacterial CFU).

## DISCUSSION

We showed a synergistic effect between avermectins and *Metarhizium* fungi on aquatic invertebrates for the first time. A similar effect was shown previously only in terrestrial insects (Colorado potato beetle, cotton moth) (*Anderson et al., 1989*; *Asi et al., 2010*; *Tomilova et al., 2016*), which are characterized by a completely different mode of fungal penetration (through the exo-skeleton). The accumulation of conidia of the fungus mainly in the gut lumen of mosquitoes coincides with studies of other researchers (*Butt et al., 2013*). However, we report the first observation of colonization of *Ae. aegypti* larvae after inoculation with *Metarhizium* conidia. It was previously suggested that only blastospores (and not conidia) are able to germinate from the gut lumen into the hemocoel of mosquito larvae (*Alkhaibari et al., 2016*; *Alkhaibari et al., 2018*). Interestingly, the larvae treated with avermectins accumulated a lower amount of conidia, but this dose was sufficient for a synergistic effect on mortality. It is likely that reduced accumulation of conidia was due to disturbance of feeding. For example, decrease in quantity of consumed food under the influence of avermectins was shown for terrestrial insects (*Akhanaev et al., 2017*).

We observed a decrease in PO activity and dopamine levels under the influence of the fungus, whereas in terrestrial arthropods, these enzymes are activated during mycoses (*Ling & Yu, 2005*; *Yassine, Kamareddine & Osta, 2012*; *Yaroslavtseva et al., 2017*; *Chertkova, Grizanova & Dubovskiy, 2018*). It has been suggested that dopamine release is associated with the general stress reactions related to the insect's responses to pathogens (*Hirashima, Sukhanova & Rauschenbach, 2000*; *Chertkova, Grizanova & Dubovskiy, 2018*). In addition, dopamine is involved in the modulation of energetic metabolism and general defense mechanisms such as phagocytosis (*Wu et al., 2015*). PO is involved in the inactivation of fungal propagules in the cuticle and hemocoel (*Butt et al., 2016*). Dopamine is involved in the PO cascade (*Andersen, 2010*); however, synchronous and unidirectional changes in the levels of PO and dopamine are not always observed during infections (E Chertkova, 2016, personal observations). Since we observed differentiation of fungal infection structures,

we suggest that some fungal metabolites inhibit the PO cascade of *Ae. aegypti* larvae. It was shown on terrestrial insects that *Metarhizium* secondary metabolites (e.g., destruxins) may reduce the number of PO-positive hemocytes (*Huxham, Lackie & McCorkindale, 1989*) and these metabolites may upregulate serine protease inhibitors, which inhibit proPO cascade (*Pal, Leger & Wu, 2007*). *Alkhaibari et al. (2018)* noted a short-term increase in PO activity in the whole-body homogenates of *Culex quinquefasciatus* larvae after infection with conidia or blastospores of *M. brunneum* Petch (4–6 h pt). It is possible that this reaction depends on species of mosquitoes as well as strain of the pathogen. Especially, inhibition of hemolymph melanization under *M. robertsii* infections was dependent from the production of secondary metabolites by different strains (*Wang et al., 2012*).

We observed activation of detoxifying enzymes (GST, EST) in *Ae. aegypti* larvae at the early stages of toxicosis and infection (12 h pt) under mono-treatments with the fungus and avermectins. However, the combined treatment led to inhibition of the activation of GST and EST. A similar effect was observed at 12 h pt for antibacterial (lysozyme-like) and acid protease activities. Combined treatment leads to either inhibition or containment of the activation of these enzymes. GST and EST are used by insects to inactivate toxic products formed by insecticide-induced toxicoses (*DeSilva et al., 1997*; *Boyer et al., 2006*; *Aponte et al., 2013*) as well as under mycosis (*Dubovskiy et al., 2012*). Especially *Tang et al. (2019)* showed that up-regulation of GSTz2 decreased the susceptibility of tephritid fruit fly *Bactrocera dorsalis* (Hendel) to abamectin. Moreover, GST may participate in inactivation of fungal secondary metabolites (*Loutelier, Cherton & Lange, 1994*) and reactive oxygen species (*Sherratt & Hayes, 2002*). Lysozyme inhibits the reproduction of Gram-positive bacteria (*Abdou et al., 2007*; *Gandhe, Janardhan & Nagaraju, 2007*; *Chapelle et al., 2009*), which (e.g., Microbacteriaceae) are among the dominant bacteria in *Ae. aegypti* larvae (*Coon et al., 2014*). It should also be noted that at 12 h pt of *Ae. aegypti* with blastospores of *M. brunneum*, a decrease in the expression levels of genes encoding defensins and cecropins (*Alkhaibari et al., 2016*), which inhibit the growth of both Gram-positive and Gram-negative bacteria and fungi, was observed (*Jozefiak & Engberg, 2017*). The inhibition of acid protease activity under combined treatment may indicate disorders in food consumption and absorption. Disruption of food absorption and starvation can increase mortality from both fungi and insecticides (*Furlong & Groden, 2003*). Thus, we assume that the physiological causes of the observed synergism lie in the initial stages of the development of infection and toxicosis.

In the later stages (48 h pt), we mainly observed activation of the enzymes (PO, GST, EST, acid proteases and lysozyme-like activity), which apparently indicates destructive processes in tissues and organs under the action of both avermectins and fungi. The increase in PO activity on the second day after treatment with avermectins was probably due to the destruction of hemocytes and the release of intracellular proPO components. We have previously shown the cytotoxic effect of avermectins on hemocytes, leading to their death (*Tomilova et al., 2016*). Additionally, the cytostatic and cytotoxic effects of the avermectins complex on various cells of warm-blooded animals are well known (*Sivkov, Yakovlev & Chashov, 1998*; *Kokoz et al., 1999*; *Korystov et al., 1999*; *Maioli et al., 2013*). Increase in PO activity under the influence of avermectins could also be symptom linked with proliferation

of bacteria (Fig. 8). The enhancement of PO is observed under development of various bacterioses and caused by damages of insect's tissues as well as by recognition of bacterial cell wall compounds, formation of hemocyte nodules and their melanization (*Bidla et al., 2009*; *Tokura et al., 2014*; *Dubovskiy et al., 2016*). An increase in GST under mycoses usually correlates with the severity of the infectious process (*Dubovskiy et al., 2012*; *Tomilova et al., 2019*) and confirms the results obtained by *Butt et al. (2013)* when studying the pathogenesis of *M. brunneum* in *Ae. aegypti* larvae. An increase in lysozyme-like activity under the action of avermectins could have occurred due to tissue destruction accompanied by the release of lysosome contents containing lysozyme (*Zachary & Hoffmann, 1984*). The activation of proteases at 48 h pt under the influence of avermectins is correlated with the increase in PO and lysozyme-like activity, which also indicates destructive changes in the tissues.

We observed an increase in the number of cultivated bacteria in the larvae when treated with both avermectins and fungi. This effect may be associated with impaired intestinal peristalsis as well as changes in the level of PO and antibacterial activity in the initial stages of toxicosis and fungal infection. Similar effects have been observed in terrestrial insects following topical infection by fungi (*Wei et al., 2017*; *Ramirez et al., 2018*; *Polenogova et al., 2019*) and are associated with the redistribution of immune responses between the cuticle and the gut. In mosquito larvae, the fungus comes into direct contact with gut microbiota, which may exhibit fungistatic properties (*Sivakumar et al., 2017*; *Zhang et al., 2018*, etc.) or, alternatively, may act as synergists of fungi, as shown by *Wei et al. (2017)* in the adults of the mosquito *Anopheles stephensi*. It is possible that conflicting data on the colonization of mosquito larvae by *Metarhizium* fungi are associated with differences in bacterial communities, which requires further research. In any case, fungi cannot successfully complete colonization in an aquatic environment, and bacterial decomposition is observed in mosquito larvae, whereas surface conidiation occurs in the air environment.

## CONCLUSIONS

In conclusion, this is the first study of the survival and physiological reactions of mosquito larvae under the combined action of avermectins and entomopathogenic fungi. The synergism observed under the combined action of these agents appears to be associated with physiological changes in the early stages of toxicosis and infection. In particular, inhibition of the activity of a number of enzymes is observed under the combined treatment associated with the detoxifying and immune systems.

Colonization of *Aedes aegypti* larvae by the fungus *Metarhizium robertsii* is shown for the first time in this study. Further investigations may be focused on studying the role of endosymbiotic mosquito bacteria in the development of toxicoses and mycoses as well as the development of preparative forms based on fungi and avermectins for mosquito control in natural conditions.

## ACKNOWLEDGEMENTS

The authors are grateful to Dr. V.A. Shilo (Karasuk Station of the ISEA) for help in organizing the experiments, to Dr. AA Alekseev for determining the exact ratio of

avermectins and their isomers in an industrial product "Phytoverm" using HPLC, to Dr. AA Miller for preparation of semi-thin sections of the mosquito larvae.

### Funding
This work was supported by the Russian Science Foundation (project No. 18-74-00090). The funders had no role in study design, data collection and analysis, decision to publish, or preparation of the manuscript.

### Grant Disclosures
The following grant information was disclosed by the authors:
Russian Science Foundation: 18-74-00090.

### Competing Interests
The authors declare there are no competing interests.

### Author Contributions
- Yuriy A. Noskov conceived and designed the experiments, performed the experiments, analyzed the data, prepared figures and/or tables, approved the final draft.
- Olga V. Polenogova performed the experiments, authored or reviewed drafts of the paper, approved the final draft.
- Olga N. Yaroslavtseva, Olga E. Belevich, Yuriy A. Yurchenko and Ekaterina A. Chertkova performed the experiments, authored or reviewed drafts of the paper, approved the final draft.
- Natalya A. Kryukova analyzed the data, authored or reviewed drafts of the paper, approved the final draft.
- Vadim Yu Kryukov conceived and designed the experiments, analyzed the data, prepared figures and/or tables, authored or reviewed drafts of the paper, approved the final draft.
- Viktor V. Glupov conceived and designed the experiments, contributed reagents/-materials/analysis tools, authored or reviewed drafts of the paper, approved the final draft.

### Data Availability
The raw data of the survival, dopamine concentration, glutathione S-transferase, nonspecific esterases, acid proteases, lysozyme-like and phenoloxidase activities are available in the Supplemental Files.

### Supplemental Information
Supplemental information for this article can be found online at http://dx.doi.org/10.7717/peerj.7931#supplemental-information.

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
