# Peer review of "Combined effect of the entomopathogenic fungus Metarhizium robertsii and avermectins on the survival and immune response of Aedes aegypti larvae"

_PeerJ, doi:10.7717/peerj.7931_

## Round 0.1 · original submission · Minor Revisions

Dear authors, please take into account all of the reviewer's concerns, especially regarding reference insertion in the methodology and data clarification requested by the reviewers.

Reviewer 1 ·

Basic reporting

The English is clear, but there is a profusion of unnecessary acronyms. Stick to general acronyms (like GST) not unusual ones like EPF- just write out entomopathogenic fungi. The amount of acronyms actually makes it harder to read, rather than simplifying it.

Experimental design

The design is fine, but there are issues in the reporting. Please see annotated PDF to highlight the portions where lack of information prevents replication. An important note is that the authors talk about activity in the haemolymph, but nowhere was haemolymph specifically prepared.

Validity of the findings

no comment

Additional comments

Generally a well written paper covering a novel topic. The reason for my decision is based largely on the fact that I want the figures changed from box and whisker plots to bar graphs, which moves it from the realm of minor to major revision. Otherwise, a worthwhile MS.

Annotated reviews are not available for download in order to protect the identity of reviewers who chose to remain anonymous.

·

Basic reporting

The manuscript is interesting and relevant. It is well structured and coherent. However, a minor changes should be addressed before its publication.
1.- Please write the genera Metarhizium always in italics! Idem with other genera or species.
2.- The first time that name an insect or microorganism, please include the author.
3.- Line 135. Replace "3rd" by "Third" because it is in the beginning of sentence.
4.- Please explain what is a "systemic colonization of Ae. aegypti". What do you mean exactly?
5.- Line 386. Replace "Alkhaibar" by "Alkhaibari"
6.- References: check insects or organisms in italics.
7.- Asterisks are not found in figure 1.

Experimental design

The experiments has been well designed and conducted, but a statistical point need to be reconsidered.
8.- "LC50" vs. "MST". Median lethal time: In a time-dependent biological assay procedure, this is the period of exposure to a pathogenic (including toxicological) stimulus which will produce death in half the test subjects. The length of exposure is a direct measure of dosage, and an increase in the period of exposure results in an increase in uptake and true dose in the same ratio. Its symbol is LT50, not to be confused with the Median Survival Time....which is the time at which death occurs in half the test subjects after exposure to a pathogenic (including toxicological) stimulus. Its symbol is MST less commonly ST50, it is not a direct measure of dosage, and it is not to be confused with the Median Lethal Time (LT50), which is a direct measure of dosage. —from An Abridged Glossary of Terms Used in Invertebrate Pathology, Third Edition). Please check this point, I think that the authors have calculated the MST.

Validity of the findings

Results match expectation but some sentences should be avoid, in particular those that report trends or tendencies (lines 288, 296, 311, 329, 354).

·

Basic reporting

The article is well written and well referenced and meets peerj standards. However, the authors in the abstract (line no 34) state, "we assume...." while it is their finding that they are stating after getting the results.

Further in line no. 115-117 where the authors end the introduction stating the aims of their study, the portion is to be rewritten with reference given to the citation Bishoff et al with year .

Experimental design

The authors have studied and reported their findings on the synergistic effect of Metarhizium robertsii and avermectins on the survival and immune response A. aegypti larvae, both which effects have been separately studied by other authors. The experimental methods are described adequately for replication.However the references to the methods adopted in finding total larval body supernatant (Line no. 156-162) and acid protease (line no 204-210) have not been given, which the authors may be asked to add.

Validity of the findings

The data sets provided and results obtained are adequate.However The data relating to 24 hours post treatment stages of toxicosis have not been provided.

---

## Round 0.2 · accepted · Accept

Thank you for your point-by-point response to the reviewer's comments. You have satisfactorily answered all their issues and inserted requested modification in the revised manuscript, therefore it is now acceptable for publication in PeerJ.